# Enhancing the Capacity and Stability by CoFe_2_O_4_ Modified g-C_3_N_4_ Composite for Lithium-Oxygen Batteries

**DOI:** 10.3390/nano11051088

**Published:** 2021-04-22

**Authors:** Xiaoya Li, Yajun Zhao, Lei Ding, Deqiang Wang, Qi Guo, Zhiwei Li, Hao Luo, Dawei Zhang, Yan Yu

**Affiliations:** 1School of Chemistry and Chemical Engineering, Hefei University of Technology, Hefei 230009, China; Sabrina_lxy@163.com (X.L.); zyj251264@163.com (Y.Z.); dlei1107@163.com (L.D.); wondreful26@163.com (D.W.); 18297919084@163.com (Q.G.); Zhiwei.Li@hfut.edu.cn (Z.L.); zhangdw@ustc.edu.cn (D.Z.); 2Department of Materials of Science and Engineering, University of Science and Technology, Hefei 230026, China

**Keywords:** Li-O_2_ batteries, composite, ORR, OER

## Abstract

As society progresses, the task of developing new green energy brooks no delay. Li-O_2_ batteries have high theoretical capacity, but are difficult to put into practical use due to problems such as high overvoltage, low charge-discharge efficiency, poor rate, and cycle performance. The development of high-efficiency catalysts to effectively solve the shortcomings of Li-O_2_ batteries is of great significance to finding a solution for energy problems. Herein, we design CoFe_2_O_4_/g-C_3_N_4_ composites, and combine the advantages of the g-C_3_N_4_ material with the spinel-type metal oxide material. The flaky structure of g-C_3_N_4_ accelerates the transportation of oxygen and lithium ions and inhibits the accumulation of CoFe_2_O_4_ particles. The CoFe_2_O_4_ materials accelerate the decomposition of Li_2_O_2_ and reduce electrode polarization in the charge–discharge reaction. When CoFe_2_O_4_/g-C_3_N_4_ composites are used as catalysts in Li-O_2_ batteries, the battery has a better discharge specific capacity of 9550 mA h g^−1^ (catalyst mass), and the cycle stability of the battery has been improved, which is stable for 85 cycles.

## 1. Introduction

In the face of huge pressure from energy conservation and emission reduction advocates, it is necessary to replace traditional fuel vehicles with electric vehicles possessing high energy capacity and long range [1,2]. The theoretical energy density of Li-O_2_ batteries is 3500 Wh kg^−1^, much higher than other batteries. Li-O_2_ batteries are expected to be used in electric vehicles and in large-scale productions [3,4,5]. However, so far, the actual specific energy of Li-O_2_ batteries reported in literature is less than half the theoretical value. This is because during the charging and discharging processes, Li-O_2_ batteries generate large polarization phenomena and high activation energy, leading to high energy loss [6]. One of the solutions to reduce energy loss for the Li-O_2_ batteries is to develop new electrocatalysts with high activity. Metallic materials, such as platinum and iridium, or their alloys, have been shown to be the best electrode materials for oxygen reduction reaction. However, the high price of metallic materials hinders their commercial application [7,8]. Hence, more and more attention has been paid to the development of electrocatalysts prepared by some non-precious metals and non-metals [9].

Recently, spinel materials (such as Co_3_O_4_) with adjustable structure and stable chemical properties have attracted the attention of researchers [10,11]. However, compared with noble-metal-based catalysts, the catalytic performance of spinel material has a big gap. Studies have found that the structural stability and performance of the spinel material can be improved by replacing Co in Co_3_O_4_ with secondary metals such as Ni, Cu and Mn [12]. Unfortunately, when choosing this material as a catalyst, as the reaction begins, particles continue to aggregate, the number of active sites decreases, and cycle performance decreases. Choosing a suitable supporting substrate to form a stable structure can effectively reduce particle agglomeration and improve catalyst activity.

Graphite carbon nitride (g-C_3_N_4_) has a graphite-like planar phase, of which the nitrogen atoms have both three-fold coordination atoms and two-fold coordination atoms. They each also contain six nitrogen lone pairs of electrons [13,14,15]. This enables the g-C_3_N_4_-based catalyst to change the electronic structure and provides an ideal position for recombination [16,17]. Therefore, when employed as an ameliorative support for the nanoparticle surface, g-C_3_N_4_ plays a significant role in the catalysis process. It can not only restrain the migration of nanoparticles, but also acts as the second active-site supplier that imparts activity to the integral catalyst [18,19]. In addition, we know that rate capability, round-trip efficiency, and cycle life in Li-O_2_ batteries are equally governed by parasitic reactions, which are now recognized to be caused by formation of the highly reactive singlet oxygen. Selected homogeneous catalyst approach to limit singlet oxygen release is a way to improve performance [20,21,22]. The uniformly dispersed particles supported on substrate are also beneficial to performance improvement due to more sufficient contact between the active constitutes and Li_2_O_2_ particles.

Taking these issues into account, we herein rationally design a scalable facile strategy for fabricating a CoFe_2_O_4_/g-C_3_N_4_ composite with the CoFe_2_O_4_ particles supported on the flaky g-C_3_N_4_. Thereinto, g-C_3_N_4_ not only provides rich catalytic sites, but also acts as a support for loading CoFe_2_O_4_ nanoparticles to restrain their aggregation. g-C_3_N_4_ and CoFe_2_O_4_ synergistically enhance the catalytic performance. The resultant CoFe_2_O_4_/g-C_3_N_4_ composite exhibits enhanced electrochemical performance of Li-O_2_ batteries with respect to discharge capacity, voltage polarization, and cycling performance when compared with single CoFe_2_O_4_. This strategy may provide an efficient and versatile approach for designing spinel-based materials as efficient Li-O_2_ batteries cathodes.

## 2. Materials and Methods

### 2.1. Synthesis

Preparation of g-C_3_N_4_: First, 4 g melamine was placed in a crucible with a cover. The crucible was annealed at 550 °C for 4 h in a muffle furnace, and the obtained product was grinded to obtain block C_3_N_4_. Next, 1 g C_3_N_4_ was dissolved in 35 mL hydrochloric acid and stirred for 30 min. The solution was then placed in a Teflon hydrothermal kettle at 110 °C for 300 min. Finally, the mixture was centrifuged, washed, and dried at 70 °C for 10 h.

Preparation of CoFe_2_O_4_/g-C_3_N_4_: First, 0.4 g g-C_3_N_4_ was dissolved in 30 mL ethylene glycol, then 0.0496 g cobalt nitrate and 0.138 g iron nitrate (n_Co_:n_Fe_ = 1:2) were added and stirred vigorously for 30 min to achieve thorough mixing. The concentrated ammonia solution was added dropwise to the mixed solution to sustain the pH at 8, which was stirred for another 30 min. Next, the mixture was put in a 50 mL Teflon-lined autoclave and kept at the temperature of 160 °C for 20 h. The products were centrifuged, washed, and dried at 70 °C for 600 min. After grinding and annealing at 350 °C for 3 h, CoFe_2_O_4_/g-C_3_N_4_ composite was obtained.

Preparation of CoFe_2_O_4_: The procedure for the preparation of pure CoFe_2_O_4_ was the same as the preparation of CoFe_2_O_4_/g-C_3_N_4,_ but without g-C_3_N_4_.

### 2.2. Material Characterization

The phase prepared in this experiment was tested with a D/MAX2500V X-ray diffractometer (XRD). The scan range was set to 10–90°, and the scan speed was 10° min^−1^. For the observation and research of the microscopic morphology of the sample, the SU8020 field emission scanning electron microscope (SEM) produced by Japan’s JOEL company was used. The surface microstructure and preliminary quantitative analysis of the sample was tested and analyzed by the JEM-2100F field emission transmission electron microscope (TEM) and energy-dispersive spectrometer (EDS). A TriStar Ⅱ 3020 V1.03 specific surface area tester was employed and the specific surface area was calculated by the BET method from the adsorption isotherm of the sample with nitrogen gas. The component elements of the sample and the analysis of element valence were performed by ESCALAB250 X-ray photoelectron spectrometer (XPS).

### 2.3. Electrochemical Performance Test

#### 2.3.1. ORR/OER Performance Test

The ORR/OER electrocatalytic performance test was carried out by an ATA-1B rotating disc electrode (RDE). Preparation of working electrode: A mixture of 10 mg CoFe_2_O_4_/g-C_3_N_4_ and 2 mg Vulcan XC-72 were added to 40 μL Nafion solution, and 2 mL isopropanol water with the given fraction (V_isopropanol_:V_Deionized water_ = 1:5) was added into the solution. The ultrasonic treatment was then used to disperse the solution, then 5 μL was pipetted and mixed suspension added to the polished glassy carbon electrode surface. Finally, the electrode was dried by natural volatilization or low-temperature drying. Once the electrode was completely dried, it proceeded to the testing process. In the measurements, glassy carbon electrode (GCE) was used as the working electrode. Saturated calomel electrode (SCE) was used as the reference electrode. Graphite electrode was used as the auxiliary electrode.

For ORR/OER polarization curve testing, the scanning speed was set to 10 mV s^−1^ and the rotation speed was 400~2000 rpm. The electrolyte was 0.1 M oxygen-saturated KOH solution. The ORR potential scanning interval was set between 0 and −0.6 V. The electrode rotation speed tested in OER was 1600 rpm, and the potential scanning interval was 0~1 V.

#### 2.3.2. Battery Performance Test

Preparation of oxygen electrode: First, 15 mg catalyst and 30 mg KB were grinded and mixed for 1 h. Next, 83.3 mg PVDF (6 wt%) and 10 drops of NMP were added to the grinded powder to form a uniform slurry without obvious particles, which was then coated on carbon paper and dried at 90 °C for 10 h.

The electrochemical performance of batteries was tested by using 2032-type coin cell. Each cell was composed of a lithium metal anode, a glass fiber separator (Whatman grade GF/D), an electrolyte containing 1 M LiCF_3_SO_3_ in TEGDME, an oxygen cathode, and two pieces of nickel foam (1 mm thick) as the filler and the current collector. This was assembled in an argon-filled glove box (M. Braun). The oxygen cathodes were prepared by coating catalyst ink onto carbon paper homogenously. The catalyst mass loading of the oxygen cathode is about 0.5 ± 0.1 mg cm^−2^.

For the evaluation of the battery discharge–charge performance and overvoltage performance test, voltage range was 2.2–4.5 V (vs. Li^+^/Li). The current density was 100 mA g^−1^.

The voltage range for the battery cycle performance test was 2.0–4.5 V (vs. Li^+^/Li). The current density was 500 mA g^−1^ and limited the capacity to 1000 mA h g^−1^.

## 3. Results and Discussions

As shown in Scheme 1, the synthesis of CoFe_2_O_4_/g-C_3_N_4_ starts with a facile hydrothermal treatment of the solution containing cobalt nitrate, iron nitrate, and the prepared g-C_3_N_4_. The phase is studied by X-ray diffraction (XRD). In Figure 1a, g-C_3_N_4_ exhibits two characteristic diffraction peaks: A peak at 27.4° is formed by the stack of g-C_3_N_4_ rings, which is the (002) crystal plane [23]; another with relatively weak intensity is at 13.0°, which is a characteristic diffraction peak of Melamine substances and mainly refers to the in-plane nitrogen pores formed by the 3-s-triazine structure [24]. By comparing the two XRD patterns, before and after alkali treatment, the peak of the material does not change, which implies that the crystal structure of g-C_3_N_4_ remains unchanged. This confirms that carbon nitride possesses good chemical stability. For the XRD patterns of CoFe_2_O_4_/g-C_3_N_4_ composite, the peaks at 18.3°, 30.7°, 35.7°, 43.3°, 57.4°, and 63.2° are CoFe_2_O_4_ diffraction peaks. The peak at 2θ position of 27.7° correspond to the typical peak of g-C_3_N_4_, implying that the CoFe_2_O_4_/g-C_3_N_4_ material has been successfully synthesized.

The surface and internal morphology of materials are observed with scanning electron microscopy (SEM) and transmission electron microscopy (TEM). Bulk g-C_3_N_4_ is composed of many layered nanosheets, which are stacked on each other (Appendix A). After alkali treatment, the g-C_3_N_4_ is stripped into flakes (Appendix A), which is conducive to the loading of spherical particles. Without g-C_3_N_4_, pure CoFe_2_O_4_ shows the particle morphology in Appendix A, while CoFe_2_O_4_/g-C_3_N_4_ exhibits stacked granular morphology connected with particles as shown in Figure 1b. TEM images indicate that CoFe_2_O_4_ nanoparticles are supported on the flaky g-C_3_N_4_ where the clear lattice fringes of 0.25 nm correspond well to the (311) plane of CoFe_2_O_4_, suggesting that these particles are CoFe_2_O_4_ (Figure 1c,d). Moreover, the element distribution C, N, O, Co, and Fe of the material is observed by energy-dispersive spectroscopic (EDS) (Figure 1e), indicating successfully prepared material of CoFe_2_O_4_/g-C_3_N_4_ composite and uniformly distributed CoFe_2_O_4_ nanoparticles on the flat g-C_3_N_4_. The specific surface area and pore volume of CoFe_2_O_4_/g-C_3_N_4_ are determined to be 244.1 m^2^ g^−1^ and 0.423 cm^3^ g^−1^ by Braunauer–Emmett–Teller (BET) analysis (Appendix A), which is much larger than those of g-C_3_N_4_. Such a big increase for specific surface area and pore volume indicates that CoFe_2_O_4_ particles supported on the flaky g-C_3_N_4_ are effective for preventing g-C_3_N_4_ from stacking on each other. The increase in specific surface area and pore size can expose more active sites, increasing the ion migration rate during charging and discharging.

The elemental composition and chemical state of synthetic materials are analyzed by X-ray photoelectron spectroscopy (XPS). The full-spectrum scan result shows that the material contains C, N, O, Fe, and Co, which proves that the CoFe_2_O_4_/g-C_3_N_4_ material was successfully synthesized (Figure 2a). In the spectrum of C 1s (Figure 2b), two strong peaks are shown at 288.1 and 284.7 eV, which respectively belong to the sp^2^ hybridized bounded carbon (C=N) and graphitic carbon (C-C) [25,26]. Figure 2c is the N 1s spectrum, which confirms that the CoFe_2_O_4_/g-C_3_N_4_ composite has three kinds of nitrogen. The 398.3 eV is the sp^2^ hybrid nitrogen (N1), the 399.2 eV is the tertiary nitrogen (N2), and the 400.9 eV is the amino functional group (N3), respectively [27,28]. The high-resolution Co 2p_3/2_ spectrum presents the signals of Co^2+^, Co^3+^, and satellite (Figure 2d), where the presence of Co^3+^, Co^2+^, and vibration satellite are respectively exhibited by the brown main peak (780.5 eV), the blue green peak at 782.2 eV, and the peak at 786.8 eV [29,30,31]. The Fe 2p_3/2_ spectrum of the catalyst also exhibits Fe^2+^, Fe^3+^, and satellite peaks, which respectively appear at 710.7 eV, 712.2 eV, and 714.1 eV [31,32,33]. The above results indicate that two-electron pairs of Fe^3+^/Fe^2+^ and Co^3+^/Co^2+^ exist in the structure of CoFe_2_O_4_/g-C_3_N_4_ material. In addition, the O1s XPS peak at 530.2 eV is the intrinsic lattice, while another peak at 531.8 eV may be the adsorbed water [34,35].

The electrochemical catalytic activity of the sample was evaluated in 0.1M KOH solution by linear sweep voltammetry (LSV). In Figure 3a, the CoFe_2_O_4_/g-C_3_N_4_ displays an onset potential (E_onset_) of 0.90 V and a half-wave potential (E_1/2_) of 0.76 V, which are more positive than those of CoFe_2_O_4_ (0.67 V) and g-C_3_N_4_ (0.65 V) catalysts (Figure 3b), signifying the higher activity for ORR. Moreover, the CoFe_2_O_4_/g-C_3_N_4_ exhibits a larger diffusion-limited current, suggesting the material has strong mass transfer ability and fast electron transfer speed. The reaction kinetics of the material are calculated and analyzed by the LSV curve at different speeds, as shown in Figure 3c. The activity of all materials at different speeds is linearly related, and the calculated number of transferred electrons show Pt/C, CoFe_2_O_4_/g-C_3_N_4_, CoFe_2_O_4_, and g-C_3_N_4_ is 4.0, 3.8, 3.4, and 1.67, respectively. The ORR reaction of CoFe_2_O_4_/g-C_3_N_4_ composite is nearly a 4e^-^ process, which indicates a better ORR catalytic performance and is close to commercial Pt/C. Furthermore, the OER curves of various catalysts show the CoFe_2_O_4_/g-C_3_N_4_ catalyst has the limiting current density of 49.3 mA cm^−2^, higher than CoFe_2_O_4_ (~18.8 mA cm^−2^), g-C_3_N_4_ (~1.93 mA cm^−2^), and Pt/C (~9.87 mA cm^−2^) (Figure 3d), indicating the CoFe_2_O_4_/g-C_3_N_4_ material has the best OER performance. This may be due to the synergies of g-C_3_N_4_ and CoFe_2_O_4_ which boost the performance.

The electrochemical catalytic ability of the material was evaluated by assembling 2032 button batteries. In Figure 4a, the CoFe_2_O_4_/g-C_3_N_4_ composite exhibits a discharge specific capacity of 9550 mA h g^−1^, significantly higher than CoFe_2_O_4_ and XC-72. The overpotential of the CoFe_2_O_4_/g-C_3_N_4_ composite as exhibited in Figure 4b is 1.21 V, which is lower than the catalysts of CoFe_2_O_4_ (1.33 V) and XC-72 (1.87 V). These results indicate that when CoFe_2_O_4_/g-C_3_N_4_ composite is used as catalyst for Li-O_2_ batteries, the degree of polarization during the discharge–charge process is relatively lower than pure CoFe_2_O_4_. Figure 4c,d show the cycle performance of CoFe_2_O_4_ and CoFe_2_O_4_/g-C_3_N_4_. The CoFe_2_O_4_/g-C_3_N_4_ composite can stably cycle 85 times, which is significantly higher than pure CoFe_2_O_4_ cathodes (16 times). In the battery cycle reaction, the flake structure of g-C_3_N_4_ can supply sufficient space to store discharge product Li_2_O_2_ and accelerate the transportation of O_2_ and Li^+^; also, its larger specific surface area and rich N content provide more reactive sites for the discharge–charge reaction. In addition, g-C_3_N_4_ provides a stable support for restraining aggregation of CoFe_2_O_4_ nanoparticles due to its high chemical stability, which leads to an increase in the stability of the composite catalyst. Therefore, from the discussion, the synergistic effect between g-C_3_N_4_ and CoFe_2_O_4_ can effectively improve the electrocatalytic activity and stability of the catalyst in Li-O_2_ batteries, which is much better than pure CoFe_2_O_4_ cathode. The capacity retention rate tests of Li-O_2_ batteries of CoFe_2_O_4_/g-C_3_N_4_ material were carried out at four current densities (Figure 4e). Compared with the first discharge capacity, the corresponding capacity retention rates of the four current densities are 67.7%, 61.3%, 48.4%, and 42.3%, respectively (Figure 4f). All results prove that CoFe_2_O_4_/g-C_3_N_4_ composite can promote ORR/OER dynamics, thus improving the rate performance of the battery. Even at a high current, the battery keeps a good capacity retention rate.

To investigate the composition and morphology changes of CoFe_2_O_4_/g-C_3_N_4_ electrodes at different stages after the discharge–charge process, Figure 5a displays XRD spectrum of batteries at different states. Compared with the catalyst in the initial state, after discharging, the surface of carbon sheet clearly exhibits characteristic peaks of Li_2_O_2_ at 33°, 35°, and 58°, and neither Li_2_O nor LiOH is detected. After charging, no diffraction peak of Li_2_O_2_ is observed, which indicates that the battery has a good reversibility. Figure 5b–d are the SEM images of the battery after different processes. From Figure 5c, the morphology of the Li_2_O_2_ after the battery being deeply discharged is clearly observed. The lithium peroxide is mainly distributed uniformly on the carbon sheet in the form of filaments. After the charging process (Figure 5d), Li_2_O_2_ disappeared completely, which is similar with the initial state (Figure 5b) and consistent with XRD results, indicating that the battery has good reversibility.

## 4. Conclusions

In short, the g-C_3_N_4_/CoFe_2_O_4_ material is synthesized by simple methods. The flake structure of the CoFe_2_O_4_/g-C_3_N_4_ catalyst accelerates the transportation of O_2_ and Li^+^ and provides sufficient space to store the discharge product of Li_2_O_2_. Both g-C_3_N_4_ and CoFe_2_O_4_ can offer catalytic sites. In addition, g-C_3_N_4_ supplies a stable support for restraining the aggregation of CoFe_2_O_4_ nanoparticles. Therefore, the Li-O_2_ batteries with such CoFe_2_O_4_ modified g-C_3_N_4_ composites as air cathodes deliver a discharge specific capacity of 9550 mA h g^−1^, and the cycle stability has been enhanced more than pure CoFe_2_O_4_ cathodes. This strategy opens opportunities for rationally exploring different modified strategies on nanostructured electrocatalysts for diverse devices.

## Data Availability

Data can be available upon request from the authors.

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
