# Peer review of "Enhancing the Capacity and Stability by CoFe2O4 Modified g-C3N4 Composite for Lithium-Oxygen Batteries"

_nanomaterials, 2021, doi:10.3390/nano11051088_

Round 1

Reviewer 1 Report

There are too many mistakes of format.  Many numbers are neither superscript nor subscripts such as “CoFe2O4/g-C3N4”.  Since material synthesis is not described quantitatively “A certain amount of”, no one can now the nature of materials.

Pore structure strongly affect the performance of Li-air battery, data of pore size distribution CoFe2O4/g-C3N4 should be added.

Reviewer 2 Report

This manuscript reports a possibile composite heterogeneous catalyst for OER/ORR in Li-O2 aprotic batteries. The work is well done a clearly presented. There are few weak points that require attention:

1) In the abstract the authors state that there is an energy crisis. This is not true at all. We are all in the middle of an energy paradigm transition but there is plenty of energy produced and available worldwide. Please correct.

2) In the abstract and throughout the text specific capacities are rpoerted in mAhg-1 without reported in respect to what mass (the catalyst mass, the electrode mass including the gas diffusion layer, etc.). Please report this clarification in the abstract and in the methodological section. Furtheremore for the sake of completeness, capacities in Li-O2 batteries are also reported in mAhcm-2 in respect to the geometric area of the positive electrode. Please update.

3) What reference electrode is isued in the ORR/OER tests in aqueous media with the RDE? is it correct that the electrolyte was an aqueous KOH solution? Please explain better the electrolyte composition in the methodological section.

4) What eis the electrolyte used in the battery test? Please report clearly in the metholodogical section.

5) What is the mass loading and overall mass of the positive electrode?

6) In the figure 2 (d-e) tentative deconvolutions of the Co and Fe detailed regions are proposed. Thiese fits are wrong. There is a very large literature concerning the complex multiplets deconvolution for each transition metal, se as an example Besinger et al Appl. Surf.Sci. 257 (2011) 2717-2730. The fits for the transition metals need a carefull check and update based on solid literatire sources.

7) In the whole text there isn't any single word about the most remarkable challenge in Li-O2 battery science: singlet oxygen release and parasitic chemistries. Please add some relevant citation  in the introduction and comment about the efficacy of the selected herogeneous catalyst approach to limit singlet O2 release.

Round 2

Reviewer 1 Report

The manuscript was much improved. Therefore, this paper is clearly worth publishing for Nanomaterials.  Accept as it is

Reviewer 2 Report

The revision accepted almost all my remarks and replied to all of them. I have still some concerns about the XPS fitting that it is too simple and not fully in agreement with the literature. However the authors cited the suggested seminal paper from Biesinger and thus readers can easily built their own opinion.

In this version the manuscript can be published